# Universal emulsion stabilization from the arrested adsorption of rough particles at liquid-liquid interfaces

Michele Zanini[1], Claudia Marschelke[2], Svetoslav E. Anachkov[3], Emanuele Marini[1], Alla Synytska[2] & Lucio Isa[1]

Surface heterogeneities, including roughness, significantly affect the adsorption, motion and interactions of particles at fluid interfaces. However, a systematic experimental study, linking surface roughness to particle wettability at a microscopic level, is currently missing. Here we synthesize a library of all-silica microparticles with uniform surface chemistry, but tuneable surface roughness and study their spontaneous adsorption at oil–water interfaces. We demonstrate that surface roughness strongly pins the particles' contact lines and arrests their adsorption in long-lived metastable positions, and we directly measure the roughness-induced interface deformations around isolated particles. Pinning imparts tremendous contact angle hysteresis, which can practically invert the particle wettability for sufficient roughness, irrespective of their chemical nature. As a unique consequence, the same rough particles stabilize both water-in-oil and oil-in-water emulsions depending on the phase they are initially dispersed in. These results both shed light on fundamental phenomena concerning particle adsorption at fluid interfaces and indicate future design rules for particle-based emulsifiers.

[1] Laboratory for Interfaces, Soft Matter and Assembly, Department of Materials, ETH Zurich, Vladimir-Prelog Weg 5, 8093 Zürich, Switzerland. [2] Department of Polymer Interfaces, Leibniz Institute of Polymer Research, Hohe Strasse 6, D-01069 Dresden, Germany. [3] Department of Chemical and Pharmaceutical Engineering, Faculty of Chemistry and Pharmacy, Sofia University, 1 James Bourchier Avenue, 1164 Sofia, Bulgaria. Correspondence and requests for materials should be addressed to L.I. (email: lucio.isa@mat.ethz.ch).

Micro- and nanoparticles adsorbed at fluid interfaces are an essential ingredient of a very broad class of materials and processes, from stabilizing emulsions and foams[1–3], to the fabrication of nanostructured materials[4,5]. Understanding the single-particle wetting properties at the interface holds the key to controlling the properties of the macroscopic systems. Young's law[6], which relates the equilibrium wetting angle of a liquid on a solid to the surface energies of the materials, is implicitly considered to apply to colloidal particles adsorbed at fluid interfaces[7]. These macroscopic assumptions break down if the surface of the colloids is heterogeneous, either chemically or topographically. Surface heterogeneities in particular have a significant role in the adsorption[8–10], motion[11,12] and interactions[13–15] of particles confined at fluid interfaces. A growing body of literature has, in fact, demonstrated that even particles with allegedly model surfaces display broad distributions[16] and ageing of their contact angles[9,10], and metastable orientations at the interface[17]. They can furthermore aggregate due to capillary multipoles[13–15] and undergo unexpected dynamics as they straddle the interface[11,12]. All these phenomena are signatures of surface heterogeneity.

Surface roughness, in principle, could be thermodynamically treated as a mere increase of the overall surface area of the particles. This has the consequence that the window of solid material contact angles for which particle adsorption at the interface is allowed, asymptotically narrows around 90° for very large particle roughness[8]. The existence of a non-monotonic dependence between surface roughness and interfacial activity of rough nanoparticles has been recently reported, with the consequence that an optimum, intermediate roughness was found to provide maximum emulsion stability[18] and adsorption efficiency[19]. Existing studies though, only link particle topography to the macroscopic behaviour of emulsions, and the investigation of particle adsorption at the microscopic level remains a significant challenge. Studying and understanding the entire adsorption process of rough particles at fluid interfaces, becomes, therefore, a necessary endeavour in a largely unchartered territory with high potential for the development of new materials.

In this work, we systematically address the sole effect of surface roughness on the wetting of colloids at oil–water interfaces. To achieve this goal, we produced model all-silica raspberry microparticles with tunable surface roughness. Smooth silica particles have been shown to reach their equilibrium positions upon adsorbing at liquid–liquid interfaces within few milliseconds[10] and to be largely free of the surface chemical heterogeneities found in polymeric particles[15,20]; silica thus constitutes an ideal choice to isolate topography from chemical effects. Akin to contact-line pinning for macroscopic droplets on structured surfaces[21], we demonstrate that surface roughness strongly pins the particle contact lines and arrests their adsorption in long-lived metastable positions. Consequently, the adsorption of rough particles induces interfacial deformations in agreement with the prediction of capillary theory[13,14]. By taking advantage of the tremendous contact angle hysteresis generated by roughness-induced pinning, we demonstrate the existence of a class of universal Pickering emulsifiers able to stabilize both oil-in-water and water-in-oil emulsions.

## Results

**Particle fabrication**. We created rough particles by electrostatically adsorbing negatively charged silica nanoparticles (12–250 nm, aka berries) onto larger silica microparticles (1–6 μm, aka cores), positively charged after surface modification. To further fine-tune their surface roughness, we used a modified Stöber process[22], in which additional silica was heterogeneously nucleated on the surface of the raspberry-like particles, filling the gaps between berries and sealing them together (Fig. 1a, more details in the Methods section). By selecting the core-to-berries size ratio and the thickness of the grown silica layer, we produced a library of particles with controlled surface roughness over a broad range (Fig. 1b,c, more details in Supplementary Table 1). We measured the particle topography by scanning individual raspberry-like particles within a dried monolayer using an AFM. For our 1 μm particles, the root-mean-squared (RMS) roughness obtained from the AFM analysis spans values from 1 (unmodified core) to 21 nm (Fig. 1d), while RMS roughness values up to 54 nm can be reached for particles with a 6 μm core (Fig. 1e). The RMS roughness of the particle surface was accurately determined by subtracting the particle curvature and can be used as a reliable parameter to describe and compare the surfaces of the different particle batches; all of them have in fact the same morphology, that is, comprising spherical asperities (more details in the Supplementary Figs 3–6).

**Single-particle contact angle measurements**. We carried out a systematic study of the wetting of our rough colloids at water/ n-decane interfaces by means of freeze-fracture shadow-casting (FreSCa) cryo-scanning electron microscopy (SEM)[16,23]. Particles spontaneously adsorbed from either the aqueous or the oil phase and were immobilized at the interface by shock freezing within less than a minute after interface preparation. After removal of the oil, we measured the particles' protrusion height in the cryo-SEM, and thus their contact angle at the interface.

In a first set of FreSCa experiments (Fig. 2a), we imaged in-situ hydrophobized 1 μm rough particles (as shown in Fig. 1b). The surface modification was achieved by means of a dichain cationic surfactant (di-$C_{10}$DMAB), already used to tailor the contact angle of silica colloids[24] (more details in the Methods and in the Supplementary Figs 1 and 2). Since the technique does not allow the direct measurement of contact angles below the metal shadowing angle (30°), we report both the average contact angles for the particles casting a shadow and the percentage of particles with a contact angle $\theta \leq 30°$ as a function of particle RMS roughness. The FreSCa analysis reveals that the native cores reach an average equilibrium contact angle of 70°, while, for the same conditions of adsorption from the aqueous phase, the rough colloids have monotonically decreasing effective contact angles for increasing RMS roughness values, as expected from the Wenzel equation[25]. Correspondingly, the percentage of particles with contact angles below 30° steadily grows, until, as displayed in the SEM image of Fig. 2a-VI, all particles with RMS roughness values of 21 nm can only breach the interface without protruding any further. Surface roughness provides, therefore, a multitude of very efficient pinning points for the three-phase contact line, and strongly arrests particle adsorption to effective contact angles that are, for hydrophilic particles adsorbing from water, significantly smaller than the ones obtained for the same smooth surface. By comparing the measured contact angles of particles with different silica layer thickness and similar RMS roughness values (for example, Fig. 2a-III and IV) and, vice versa, similar silica thickness and very different surface roughness (for example, Fig. 2a-II and VI), we conclude that surface topography is the factor having the dominant role in defining the particles' contact angles (see Supplementary Table 1 in the SI for additional details). Furthermore, pH-dependent electrophoretic mobility measurements showed that particles subjected to different degrees of smoothening present similar surface chemistries, excluding any major role of possible chemical heterogeneities caused by residual polyelectrolyte exposed at the silica surface (see Methods and Supplementary Fig. 2).

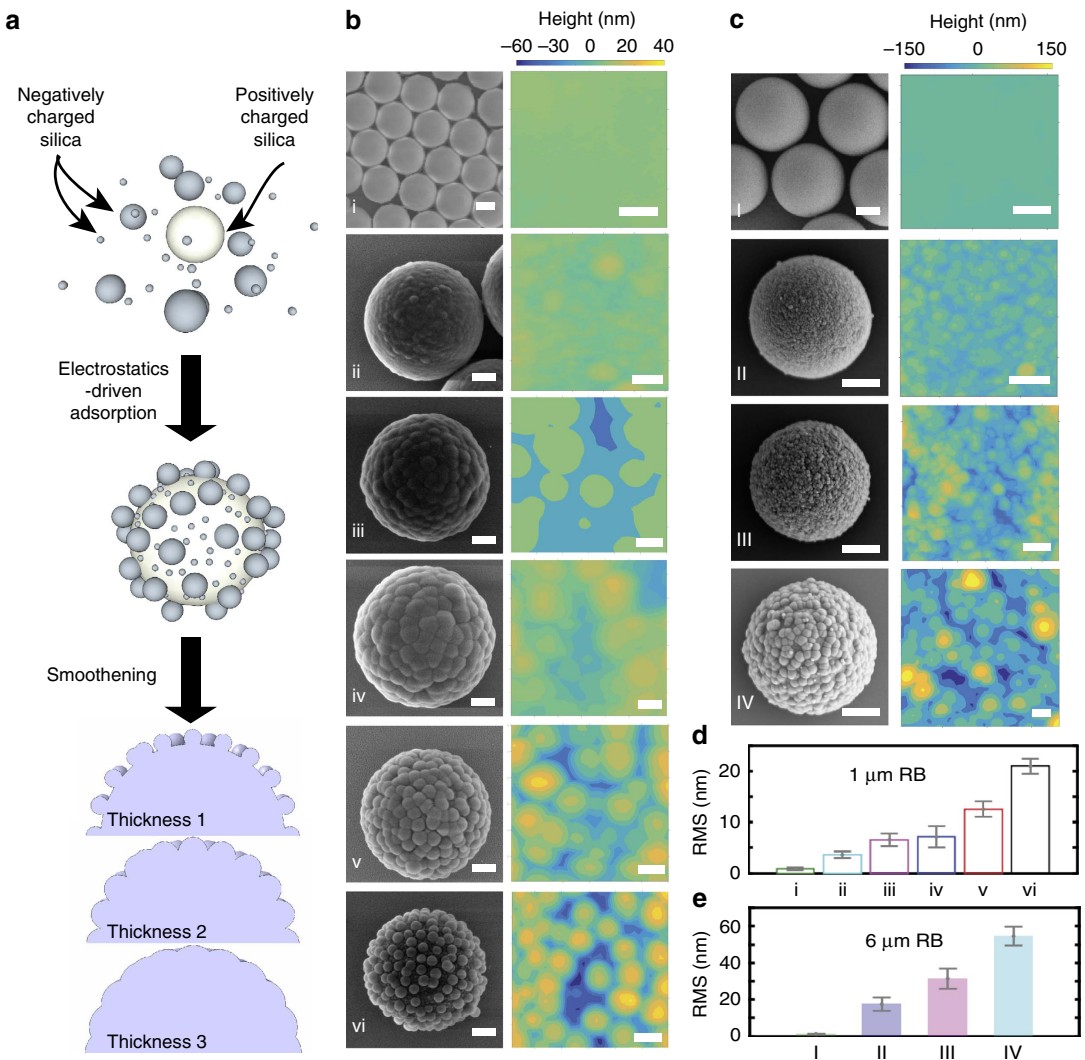

**Figure 1 | Fabrication and characterization of model rough particles. (a)** Schematics of the fabrication process. In white, the modified, positively charged silica core. In grey, the silica berries. Roughness tailoring takes place via heterogeneous nucleation of silica. The thickness of the silica layer, together with the core-to-berries size ratio, determines the final surface roughness. **(b)** SEM micrographs of the library of 1 μm rough particles paired with the respective AFM scans of the particle surface after removing the underlying curvature. The scale bar for the SEM images is 200 nm, while for the AFM is either 50 nm (i) or 200 nm (ii–vi). **(c)** SEM micrographs of the library of 6 μm rough particles paired with the respective AFM scans of the particle surface after removing the underlying curvature. The scale bar for the SEM images is 2 μm, for the AFM is 500 nm. **(d,e)** RMS roughness for the different classes of fabricated particles. Each bar represents the mean value and the error bars indicate the s.d. extracted from the data distributions.

In a second set of experiments (Fig. 2b), we chose four different rough 1 μm raspberry-like particles out of the produced library and we covalently modified them by means of a bromo-silane coating to obtain a nearly neutrally wetting surface, $\theta \approx 90°$ (see Methods section and Supplementary Table 2). Following this, the particles could be dispersed in both the polar and non-polar phase, without marked aggregation, allowing the unique opportunity to study the spontaneous adsorption of the exact same objects from both water and oil. The data show that a very small contact angle hysteresis is seen for the smooth particles ($\approx 10°$), but that the difference between the measured contact angles after adsorption from the two phases grows tremendously for increasing surface roughness, reaching contact angle hystereses of up to 100° for the roughest particles. Consequently, the rough particles adsorbing from the water are effectively more hydrophilic and the colloids adsorbing from the oil are more hydrophobic for the same surface chemistry. This is due to surface-roughness-induced pinning and arrest of the contact line during adsorption.

The final proof of this fact can be obtained in a third set of experiments (Fig. 2c), where chemically unmodified rough particles were used. By taking advantage of the fact that for 6 μm particles sedimentation is faster than aggregation (see Supplementary Note 9), they were successfully dispersed in $n$-decane, despite being hydrophilic, and were let sediment and adsorb to the water/$n$-decane interface. The smooth particles with RMS roughness of 1.2 nm cross the interface and reach the expected contact angle for pristine silica surfaces (25°), while the rough particles stop their protrusion into the aqueous phase at earlier stages. The marked surface hydrophilicity alone cannot overcome the contact-line friction imposed by the surface topography. Therefore, particles that are highly hydrophilic in nature, but adsorb from the oil phase, become effectively more hydrophobic, and for an RMS surface roughness of 54 nm an apparent contact angle of $\approx 140°$ is measured.

**Interfacial deformations.** All tested particles therefore present a surface-roughness-dependent arrest of their adsorption.

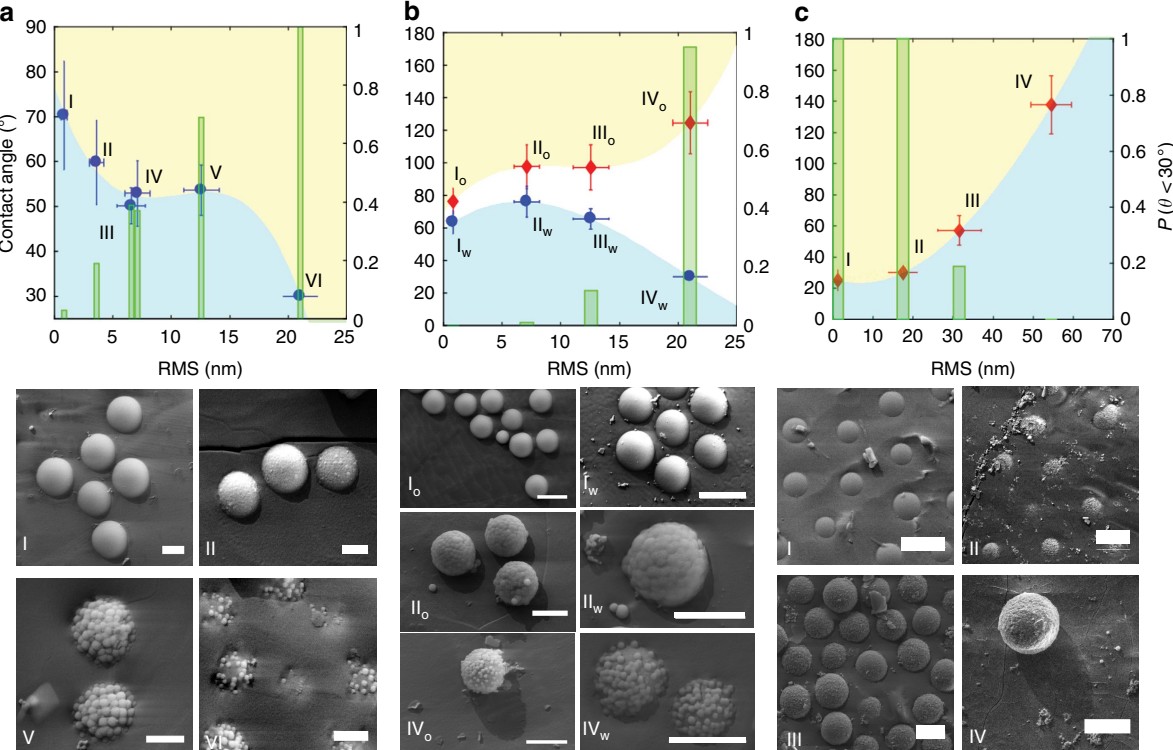

**Figure 2 | Effect of surface roughness on the wetting of colloids.** Measured contact angles as a function of particle's RMS surface roughness from FreSCa cryo-SEM images. Blue circles and red diamonds indicate spontaneous adsorption from the aqueous phase and from the oil phase, respectively. Each point represents the mean value and the error bars indicate for the s.d. extracted from the data distributions. The green bars represent the percentage of measured particles with a contact angle $<30°$ (right $y$ axis). (**a**) Colloids spontaneously adsorbed from the aqueous phase in the presence of 0.1 mM di-C10-DMAB. The scale bar for the SEM micrographs is 500 nm. (**b**) Surface-roughness-induced wetting hysteresis obtained for bromo-silane-modified rough particles. The FreSCa images $I_o$–$IV_o$ correspond to particle adsorbing from the oil, while in the images $I_w$–$IV_w$ the particles adsorb from the water. The scale bar for the SEM micrographs is 1 μm. (**c**) Spontaneous adsorption of unmodified rough silica particles (6 μm core size) reaching the interface upon sedimentation in the oil phase. The scale bar for the SEM micrographs is 5 μm.

To confirm that this is due to the pinning of the contact line on surface asperities, we carried out additional measurements aimed at carefully visualizing the contact line as well as the surrounding interface. Colloids with heterogeneous surfaces that pin the contact line are indeed predicted to cause interfacial deformations, which have a quadrupolar symmetry in the far field[13,14]. The latter have never been directly observed for rough spherical microparticles. Here the interface was imaged by using a modified version of the gel-trapping-technique (GTT)[26] (see the Methods section). In these experiments, both smooth and raspberry-like particles of 1 μm diameter were embedded in an epoxy replica of the interface, following their spontaneous adsorption from the oil phase. Figure 3a–d display AFM scans of isolated particles on the GTT replicas, showing the interface and the portion of the particles originally exposed to water. The contact line was automatically detected by a custom image-analysis algorithm (see the Methods section) and is overlaid onto the AFM scans. The qualitative perception that rougher particles possess more irregular contact lines is quantitatively confirmed in Fig. 3e. Here for each particle from Fig. 3a–d, the two-dimensional projections of the three-phase contact line on the interface plane are plotted as function of the central (azimuthal) angle $\varphi$. In-plane contact line corrugations are also accompanied by vertical deformations of the interface. Figure 3f,g display the interface height profile around the particles at two specific radial distances from the particles' edge ($r$), showing oscillations whose amplitude decreases moving away from the particle. A more detailed analysis reveals that the amplitude of the interface deformation decays as $r^{-2}$ in the far field (see Supplementary Fig. 8),

as expected for capillary quadrupoles[14]. The far-field quadrupolar symmetry can also be clearly recognized in the interface reconstruction displayed in Fig. 3h (for the discretized three-dimensional (3D) reconstruction of the interface profile around a rough particle trapped at a w/o interface see Supplementary Fig. 7). Close to the particle edge, higher-order deformations are visible, but they rapidly decay away from the particle. The control measurements on a smooth hydrophobic particle with uniform surface chemistry show a circular contact line and a flat interface (more details in Supplementary Fig. 8).

**Pickering emulsions.** In addition to the fundamental understanding of roughness effects on particle adsorption and interface deformations, the results reported above have significant impact on the stabilization of Pickering emulsions. In fact, the commonly accepted rule is that hydrophilic particles stabilize oil-in-water (o/w) emulsions, while hydrophobic ones stabilize water-in-oil emulsions (w/o)[3]. However, the data in Fig. 2 demonstrate that rough particles with a surface functionalization for which smooth analogues are hydrophilic, can behave as if they were very hydrophobic when adsorbed from the oil phase. This leads to the striking fact that the exact same rough particles can be used to stabilize both o/w and w/o emulsions. Indeed, Fig. 4 shows that emulsifying 50:50 water:$n$-decane mixtures in the presence of the bromo-silane-modified 1 μm rough particles (RMS = 21 nm) yields o/w emulsions if the particles are initially dispersed in water (Fig. 4a), while dispersing the exact same particles in the oil leads to the stabilization of w/o emulsions (Fig. 4b). Details of the emulsification procedure are given in the Methods section.

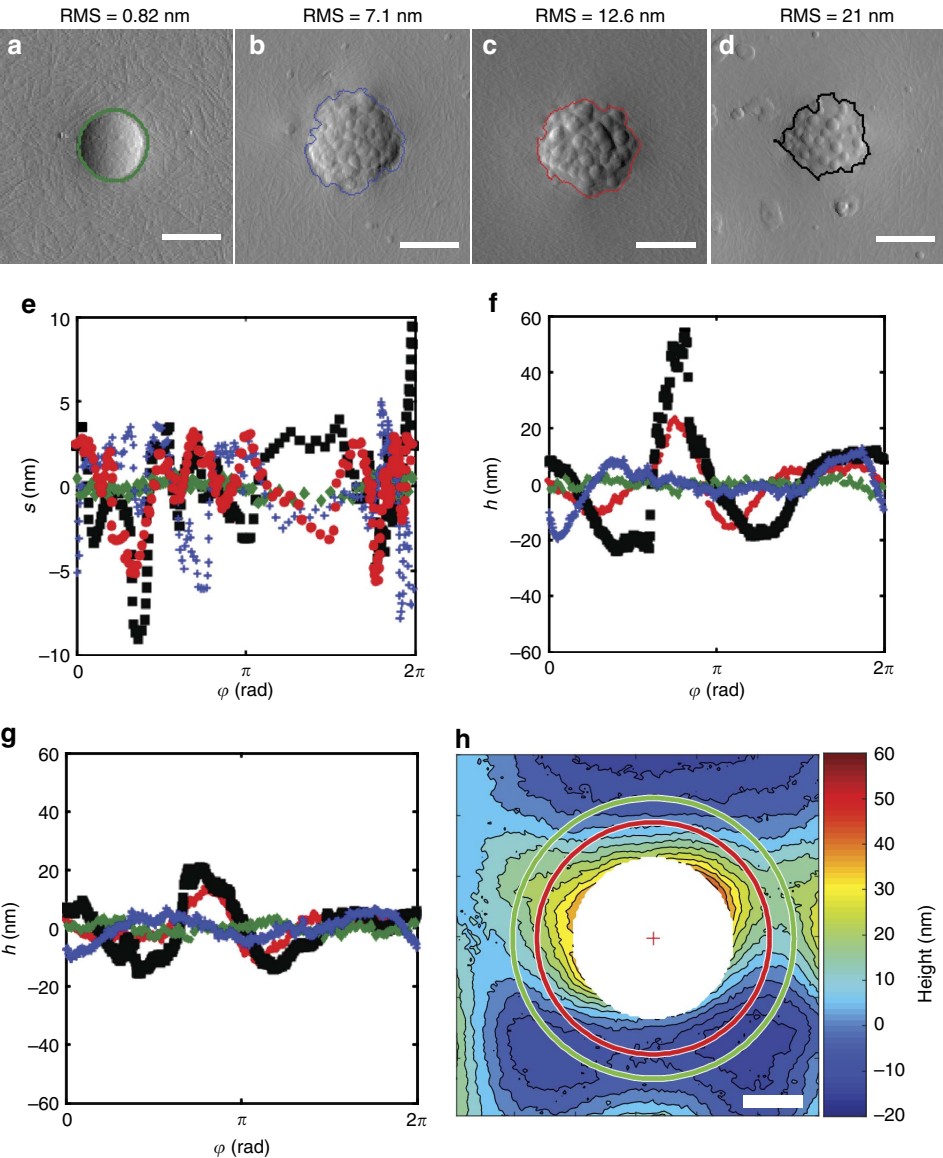

**Figure 3 | AFM study of roughness-induced interfacial deformation.** (**a–d**) AFM images (error channel) of GTT replicas of isolated particles adsorbed onto a water/*n*-decane interface from the oil side. The particles' edges are highlighted. The scale bar is 500 nm. (**e**) In-plane oscillations (*s*) of the three-phase contact line for four different surface roughness values. Green diamonds: OTS-hydrophobized smooth particle (RMS = 0.82 nm). Blue crosses: rough particle with RMS = 7.1 nm. Red dots: rough particle with RMS = 12.6 nm. Black squares: rough particle with RMS = 21 nm. All rough particles have a bromo-silane surface coating. (**f,g**) Out-of-plane interface deformations (*h*) measured around the individual particles (**a–d**) at constant radial distances from the particle's edge of 300 nm and 500 nm, respectively. The same labelling as for **e** is applied. (**h**) Height contour plot of the interface around a rough particle with RMS roughness of 12.6 nm, showing the global quadrupolar symmetry of the interface deformation. The scale bar is 500 nm. The red and green circles represent distances of 300 and 500 nm from the particle edge, respectively.

Smooth particles with the same surface chemistry, corresponding to a 65–75° contact angle, instead follow the expected route and are only able to stabilize o/w emulsions, irrespective of the medium in which they were suspended before emulsification (Fig. 4c,d). These results directly demonstrate that universal particle emulsifiers for both o/w and w/o emulsions can be obtained using objects with large contact angle hysteresis that are irreversibly trapped into metastable wetting states during adsorption. Our observations confirm and expand earlier predictions of the link between contact angle hysteresis and dual emulsion stabilization[27]. We finally remark that the possibility to disperse the particles in the two solvents is important and that their surface chemistry needs to be appropriately chosen, that is, to be close to neutrally wetting.

## Discussion

In this work, we have provided a comprehensive analysis of the role of surface roughness on the adsorption of colloids at fluid interfaces. After fabricating all-silica rough microparticles with tunable surface roughness and controlled surface chemistry, we investigated their adsorption and wetting at oil–water interfaces at the single-particle level using a range of complementary techniques. The main outcome of our experiments is the direct observation of the fact that surface roughness causes pinning of the contact line and consequently arrests particle adsorption in metastable positions (see Supplementary Fig. 9), which can be as far from equilibrium as effectively turning hydrophilic particles into highly hydrophobic ones. Associated to contact-line pinning, the interface around the particles deforms. Increasing surface

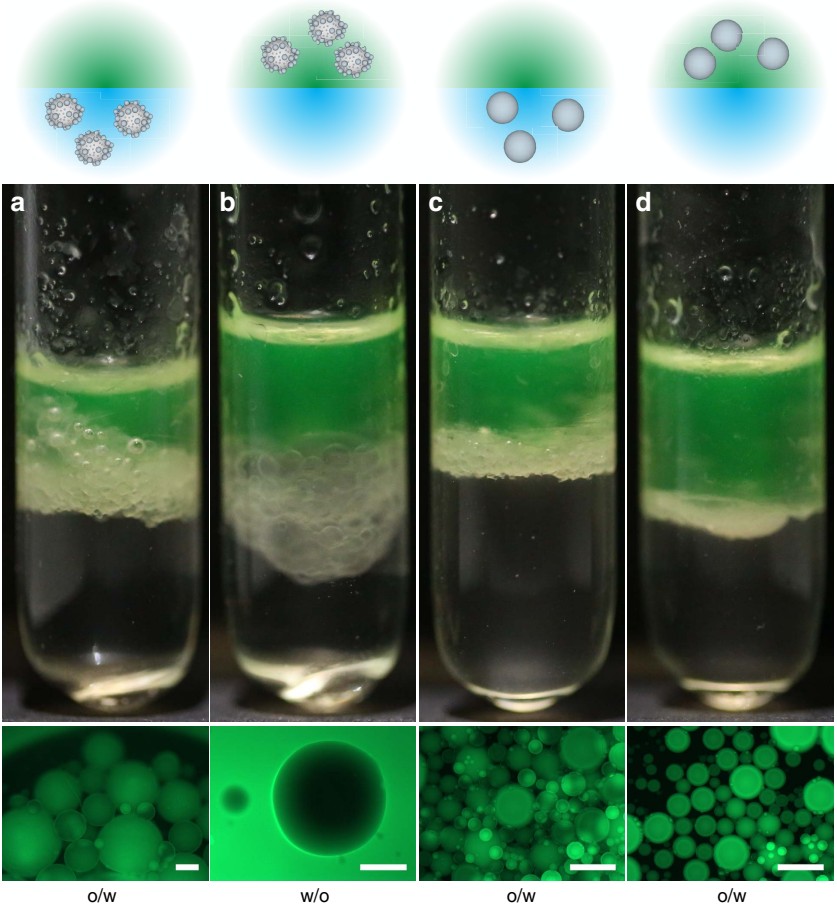

**Figure 4 | Emulsions stabilized by rough and smooth particles.** Pickering emulsions stabilized by rough particles (**a,b**) and smooth particles (**c,d**). In (**a,c**), the particles are initially dispersed in water, while in (**b,d**), the particles are initially suspended in fluorescently labelled *n*-decane. All particles were surface-modified with the bromo-silane coating. The scale bar is 200 μm.

roughness makes these phenomena more pronounced. We remark that the range of surface heterogeneities for our particles are on a different scale than surface defects previously studied[28], for which thermally activated contact line motion over the defects has been observed[9]. In fact, the energy barriers associated to the contact line moving over single asperities on our rough particles can be directly calculated[29] and are at least of the order of $10^3 k_BT$ (see Supplementary Note 8). This implies that our rough particles are strongly trapped in metastable states and even the energy introduced during an emulsification process is not sufficient to induce a relaxation of the particles' position relative to the interface. The large energy barriers associated to the motion of the contact line over the surface asperities are, therefore, responsible for the observed contact angle hysteresis. We remark here that heterogeneous wetting, for example, the creation of isolated pockets of water on the surface of a hydrophilic particle breaching into the oil phase, could also lead to contact angle hysteresis, but it was never experimentally observed in our surface roughness range.

These considerations define future design criteria to produce universal particle-based emulsifiers. They should be dispersible in both polar and non-polar fluids and have a surface roughness capable to impart large contact angle hysteresis. Controlling the interfacial deformations associated to roughness can additionally be a tool to tune capillary attractive forces, which can be used as an effective way to improve the interfacial rheology of droplets[30] and obtain emulsions with higher mechanical stability. Despite

synthetic efforts to produce model particles with controlled surfaces, a great range of naturally occurring particulate systems, such as spores or starch granules, exhibit *per se* rough and heterogeneous surfaces[31]. Therefore, prospective emulsifiers and the inspiration for their design may be directly taken from nature.

## Methods

**Materials.** *n*-decane (>99%, ABCR GmbH), Didecyldimethylammonium bromide (di-C$_{10}$-DMAB, >98%, TCI), Hydrogen peroxide (H$_2$O$_2$, VWR, 30%), Ethanol (absolute, Merck), tetraethyl orthosilicate (TEOS, Sigma-Aldrich), ammonia solution (NH$_4$OH, 25%, Merck), polydiallyldimethylammonium chloride (Poly-DADMAC, 400–500 kDa, 20 wt%, Sigma-Aldrich), silica nanoparticles 72 nm ± 6 nm (Klebosol, Clariant, Switzerland), silica nanoparticles 39 nm ± 4 nm (Klebosol, Clariant, Switzerland), silica nanoparticles 12 nm (CV<15%, Corpuscolar Inc, USA), octadecyltrichlorosilane (OTS, 95%, ABCR GmbH), decaline (Decahydronaphtalene, mix cis + trans, anhydrous, Sigma-Aldrich), Norland optical adhesive (NOA63, Norland products, USA), 3-aminopropyl-triethoxysilane (APTES, 97%, ABCR), α-bromoisobutyryl bromide (BrIn, 98%, Aldrich), triethylamine (99%, Sigma-Aldrich), dichloromethane (anhydrous, 99.8%, Acros Organics), dichloromethane (99.99%, Acros Organics), 6.27 μm silica microparticles (SiO$_2$-R-10416, Microparticles, Germany). All products were used as received. All reported particle sizes are nominal diameters.

**Fabrication of raspberry-like particles.** Silica nanospheres (berries) were anchored electrostatically onto larger, positively modified silica colloids (cores)[32–34]. In a typical procedure, homemade Stöber silica particles[35,36] (0.11 wt%, $d = 960$ nm ± 36 nm) were modified in a 0.5 wt% aqueous solution of poly-DADMAC (6 ml) to render their surfaces positively charged. The suspension was then washed three times with MilliQ water to remove the unbound excess of polyelectrolyte. Afterwards, the nanoparticle suspension, containing the berries,

was added to the positively modified core colloids (0.11 wt% suspension). 72 nm (1 wt%), 39 nm (1 wt%) or 12 nm (5 wt%) silica nanoparticles (berries) were successfully adsorbed onto the surface of the cores during 30 min of vigorous stirring intercalated by an ultra-sonication step of 10 min (Bandelin Sonorex RK 31, Ultraschallbad, Switzerland). The minimal amount of silica berries, needed to fully cover the cores' surface, was estimated by assuming that the berries form a close-packed crystalline monolayer onto the cores. We made sure that the small colloids were in excess, thus guaranteeing a more uniform surface coverage. Following the adsorption of either 72- or 39-nm-berries, 12 nm SiO$_2$ particles were also added in a second step. This ensured that the 12 nm particles adsorbed into the available interstitial positions between the larger and already adsorbed berries, leading to a more uniform distribution of nucleation sites for the further growth of a smoothening silica layer. The same process was applied for the fabrication of larger raspberry-like particles. In this case, 6.27 µm silica core-particles (0.33 wt%) were positively modified in a 6.6 wt% aqueous solution of the polyelectrolyte. After washing the suspension several times, 72 nm commercial silica particles or homemade polydisperse silica particles with an average hydrodynamic radius of 250 nm were electrostatically adsorbed on the surface of the positively modified cores.

In order to tune the surface roughness, a modified Stöber process[22] was used to grow an additional silica layer on the electrostatically coupled raspberry-like particles. A 5 vol% TEOS solution in EtOH (Supplementary Table 1) was added to a mixture of EtOH/NH$_4$OH/H$_2$O (77:13:10 vol%, 0.001 wt/vol%) containing 0.011 wt% of raspberry-like particles at an injection rate of 2 ml per hour for the smaller raspberry-like particles and 0.1 ml per hour for the larger raspberry-like particles, respectively. Both the injected volume and the injection rate affect the final thickness of the silica layer. When a rate of 2 ml per hour was used, the injection of the silica precursor was split into periods of 10 min each, followed by 30 min of equilibration time to let the mixture homogenize and ensure that all the TEOS was used before the new addition. This sequence reduces the formation of secondary nuclei by homogenous nucleation, thus leading to a more controlled output. The smoothening reaction was carried out under continuous ultra-sonication to prevent cluster formation and particle aggregation during the growth of the silica layer.

**In-situ hydrophobization by cataionic surfactant adsorption.** *In-situ* hydrophobization was carried out by means of a dichain cationic surfactant in water. Typically, 200 µl of rough and smooth silica particles (1 wt%) were dispersed in a 0.1 mM solution of the dichain cationic surfactant (di-C$_{10}$DMAB) following an established hydrophobization procedure from the literature[24]. As described by Binks and co-authors[24], spontaneous particle adsorption takes place at the fluid interface between *n*-decane and the used aqueous surfactant solution. To ensure that all colloids could experience the same adsorption conditions, each suspension was centrifuged and washed several times with the surfactant solution until the surface tension of the suspension's supernatant was the same as the surface tension of the surfactant solution used in the washing step. The surface tension of the surfactant solutions against *n*-decane was measured using pendant drop tensiometry (DSA100, Krüss, Germany). The measurements were performed in a quartz cuvette, where a 6 µl aqueous droplet was formed in *n*-decane. The Langmuir isotherm was extracted varying the surfactant concentration from 0.01 mM up to 1 mM and the working conditions for the FreSCa experiments are marked by the green circle in Supplementary Fig. 1. We additionally verified that particles with different smoothening had an analogous surface chemistry by performing pH-dependent zeta potential measurements before and after surfactant adsorption (instrument and procedure details below and data shown in Supplementary Fig. 2)

**Covalent surface modification by bromo-silanes.** As-produced raspberry-like particles were surface-activated in a mixture of H$_2$O$_2$/NH$_4$OH/H$_2$O (volume ratio of 1:1:1, 3.3 wt/vol%) at 70 °C for 10 min. In this way, a uniform layer of silanol groups was created on the particles' surface enhancing the homogeneity and efficiency of the subsequent silanization step. The activated particles were purified by centrifugation/re-dispersion in deionized water, and dried under reduced pressure at 60 °C. Afterwards, the particles were stirred for 48 h in a 3 wt% APTES solution in deionized water to introduce amino groups to the surface. The APTES-modified particles were then purified by repeated washing and centrifugation cycles in ethanol and dried in a vacuum oven at 60 °C. To confirm the quality of the particles' surface modification, zeta-potential measurements were carried out[37]. The pH-dependent zeta potential of the particles in dispersion was measured using a Zetasizer Nano ZS (Malvern Instruments, UK) and an MPT-2 autotitrator. The particles were suspended in a 10$^{-3}$ M KCl aqueous solution to obtain a slightly opalescent dispersion, for which the Debye radius is much smaller than the particle radius. The pH was automatically adjusted by adding 0.1 M KOH or HCl aqueous solutions. Three measurements were recorded for each sample at each pH value and the Smoluchowski model was applied to extract the zeta potential. The surface-activated particles have their isoelectric point at pH = 2.7, while the APTES-modified colloids present their isoelectric point at pH = 6.4.

The final functionalization was achieved by grafting α-bromoisobutyryl bromide (0.2 ml, 1.6 mM) onto the aminosilane-modified particles (1 wt%) by incubation in dry dichloromethane (5 ml) in the presence of triethylamine (0.4 ml)

at room temperature for 4 h. The particles were then purified by centrifugation/re-dispersion in dichloromethane, water and ethanol, and eventually dried under vacuum at 60 °C for harvesting.

**OTS functionalization.** Homemade smooth silica particles with an average diameter of approximately 1 µm were hydrophobized with OTS[38]. An aqueous suspension (containing 10 mg of particles) was dried at 60° in a vacuum oven and then the particles were re-dispersed in anhydrous decaline (5 ml); 30 µl of OTS were then added. The hydrophobization reaction took place for 4 h at a constant temperature of 25 °C. The modified particles were purified several times by centrifugation/re-dispersion in ethanol, dried at 70 °C and subsequently dispersed in *n*-decane. The OTS-modified particle displayed in Fig. 3a shows a protrusion height in the aqueous phase of 112 ± 4 nm (measured with AFM), corresponding to a contact angle of 141°, assuming a particle radius of 500 nm.

**Macroscopic contact angle measurements.** Static and dynamic contact angle measurements of water droplets on flat modified substrates (mimicking the respective particle modifications) were performed by the sessile drop method using the drop shape analysis technique (OCA35, DataPhysics Instruments GmbH, Germany and DSA100, Krüss, Germany) in order to serve as a reference to the contact angle measurements for the single particles. Fresh deionized water was used for all measurements. For advancing contact angle measurements, a 10 µl water droplet was formed gradually onto the sample surface at a flow rate of 0.25 µl s$^{-1}$. Receding contact angle values were recorded as the droplet volume was reduced at a flow rate of 0.25 µl s$^{-1}$. Both static and dynamic measurements were performed in air and in *n*-decane (see Supplementary Table 2).

**Freeze-fracture shadow-casting (FreSCa) cryo-SEM.** Sample preparation for the FreSCa cryo-SEM measurements was carried out following the standard procedure[16,23].

For the experiments where the particles adsorbed from water, 0.5 µl of the aqueous particle suspensions were pipetted in a homemade copper holder, pre-cleaned and hydrophilized, covered by 3.5 µl of *n*-decane and then shock-frozen with liquid propane jets (Bal-Tec/Leica JFD 030, Balzers/Vienna).

When the particles were adsorbed from the oil phase, 0.5 µl of milliQ water were injected in the hydrophilic copper holder and subsequently covered by 3.5 µl of *n*-decane containing the particles. The samples were then shock-frozen with liquid propane jets (Bal-Tec/Leica JFD 030, Balzers/Vienna).

The shock-freezing hinders water crystallization during solidification, leading to negligible thermal expansion and very strong adhesion of the particles to the vitrified water, even for large contact angles[23]. The frozen samples were fractured in high-vacuum conditions (10$^{-6}$ mbar) and cryogenic temperatures (−120 °C) in a freeze-etching device (Bal-Tec/Leica BAF060 device). Then, the samples were freeze-dried for 1 min at −100 °C and coated with 3 nm tungsten at a deposition angle of 30° followed by additional 3 nm tungsten at continuously varying angles between 30° and 90°. This procedure guarantees a uniform metallic coating to achieve high magnifications. The freeze-fractured, metal-coated samples were then transfered in a pre-cooled SEM (−120 °C) (Zeiss Gemini 1,530, Oberkochen) for imaging.

For the used coating conditions, only particles protruding through the interface with a contact angle greater than 30° cast shadows. Single-particle contact angles can be reliably measured only for particles casting a shadow. In principle, contact angles can also be determined for particles that do not cast a shadow, by measuring their cross-section at the interface, but in this case an assumption on the particle diameter (buried under the interface and thus invisible) has to be made[23]. In order to avoid errors arising from particle size polydispersity, we simply assigned an angle θ ≤ 30° to these particles and counted their fraction relative to the total number of imaged particles. The results can be found in Fig. 2. The reported data are the averages and s.d.s of the contact angle distributions typically measured over roughly 100 particles.

**AFM roughness analysis.** To characterize the particle surface roughness, individual colloids were scanned in tapping-mode by means of an AFM (JPK Nanowizard3, JPK, Germany), where each colloid was within a dried particle monolayer produced by convective assembly[39]. Imaging was conducted with silicon nitride cantilevers (OMCL-AC160TS, Olympus microcantilevers, Japan) having a spring constant of about 40 N m$^{-1}$ and a resonance frequency of about 300 kHz in air under ambient conditions. Additional details of the AFM roughness analysis and the comparison between experimental and simulated roughness data can be found in Supplementary Note 5 and in Supplementary Table 3, respectively.

**Characterization of interfacial deformations.** To reconstruct the interfacial deformations in the presence of adsorbed colloids at the liquid–liquid interface, a modified version of the GTT was used[11,26]. Particles dispersed in *n*-decane were injected in the apolar phase and let sediment to the hot (70 °C) 2-wt%-gellan-mixture/*n*-decane interface. After 15 min of equilibration time, the samples were slowly cooled down to immobilize the particles at the interface. After removal of the oil, ultraviolet-curable glue (Norland NOA63) was used to replicate the interface.

Since the replica shows the particles adsorbed at the interface viewed from the water side, in order to visualize the particle contact line, we could only scan hydrophobic particles (particle equator buried into the replica). For this reason, we chose to image the bromo-silanized raspberry-like particles adsorbed from the oil, which show apparent contact angles larger than 90°. Smooth particles with the same surface functionalization can, therefore, not be used in these experiments. In order to provide a control, we used instead smooth OTS-functionalized spheres adsorbed from the oil.

Isolated trapped colloids were imaged using an AFM along two orthogonal scanning directions to exclude image artifacts. Tapping-mode imaging was used with silicon nitride cantilevers (OMCL-AC160TS -Olympus microcantilevers, Japan, $k \sim 40\,\text{N m}^{-1}$ and resonance frequency of $\sim 300\,\text{kHz}$) in air under ambient conditions. From each scanning direction, two sets of images were obtained, namely the trace and retrace signals. In this way, we recorded four images of the same region of interest. Thermal drift occurring during the AFM acquisition along the orthogonal directions was corrected by finding the maximum two-dimensional correlation among the scans and by shifting them accordingly with respect to a common reference. The four drift-corrected images were overlaid to improve the signal-to-noise ratio. A custom-written edge detection routine based on Canny[40] and Sobel[41] method was applied to locate the position of the three-phase contact line (MATLAB and Statistics Toolbox Released 2016b, T.M., Inc., Natick, Massachusset, US). The most accurate detection of the particle edge is obtained by applying the algorithm to either the phase or the error images of the AFM, which are both sensitive to material contrast. The coordinates of the detected edge were then used for the 3D reconstruction of the contact line profile, after removing any background tilt. The 3D profile of the interface around the particle was then discretized in a series of concentric circles, spaced by either 25 or 50 nm (approximately 2–4 pixels), and starting at the smallest circle inscribing the contact line to allow for one-dimensional representations of the interface profile in polar coordinates (see Supplementary Fig. 7).

**Emulsification.** The bromo-silanized particles were dispersed both in water and in fluorescently labelled n-decane via intense ultra-sonication. The oil-soluble fluorescent dye BODIPY 493/503 (ref. 42) was used to improve the contrast in the optical images and to visualize directly the type of emulsion in a fluorescence microscope. Two hundred microlitres of a 2 wt% particle suspension were emulsified with the same amount of the second immiscible phase (1:1 oil-to-water ratio) by means of a miniemulsifier (T10 ULTRA-TURRAX, Ika, Germany). The emulsification process lasted 30 s at 11,000 r.p.m. and was carried out in an Eppendorf tube. The produced emulsions were transferred to elongated glass vials for better visualization. Fluorescence images were acquired with an Axio Observer D1 (Axioscope, Zeiss, Germany) after placing a small amount of the emulsion between two glass coverslips.

**Data availability.** The data that support the findings of this study are available from the corresponding author upon reasonable request.

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

## Acknowledgements

We acknowledge Jan Vermant for helpful discussions, Nicholas D. Spencer and André Studart for access to instrumentation and W. Woigt and K. Masania for assistance with the schematic 3D rendering of the smoothening process. L.I. and M.Z. acknowledge financial support from the Swiss National Science Foundation grant PP00P2_144646/1. L.I. and S.A. acknowledge financial support from the Scientific Exchange Programme

NMS.CH grant Sciex 14.082. S.A. acknowledges financial support from the Horizon 2020 project ID: 692146-H2020-eu.4.b 'Materials Networking'. M.Z. and L.I. acknowledge the ETH Scientific Center for Optical and Electron Microscopy (ScopeM) for technical support. A.S. and C.M. acknowledge financial support from the Deutsche Forschungsgemeinschaft (DFG) grant SY 125/4-1.

## Author contributions

L.I. and M.Z. devised and defined the study. M.Z. and E.M. synthesized the raspberry-like particles. C.M. and A.S. carried out the surface modification with bromo-silanes. S.A. wrote the roughness measurement codes for AFM image analysis. M.Z. performed all microscopy and emulsification experiments. All authors analysed and discussed the data and contributed to the writing of the manuscript.

## Additional information

**Competing interests:** The authors declare no competing financial interests.

**Publisher's note**: 

