## [Peer Review File · Nature Communications]

Reviewers' comments:

Reviewer #1 (Remarks to the Author):

The authors have shown how the surface roughness of a particle and not just its chemical composition can effect the adsorption of said particles at the oil/water interface. I find this manuscript well written and in my opinion the conclusions are well supported by their data.

However, I do not see the novelty of the work over and above that already published in the literature and in order to warrant acceptance into Nature Communications. Behrens and co-workers (ref 21) and Mable et al (ref 22) have already conducted extensive studies on how the surface roughness effects Pickering emulsifier performance. Therefore I cannot recommend publication.

Reviewer #2 (Remarks to the Author):

The manuscript by Zanini et al. describes a thorough study, in which model silica particles with tunable roughness and identical surface chemistry were used to evaluate roughness effects on the particle contact angle with an oil-water interface. The results confirm a previously predicted, but not experimentally confirmed pinning of the three phase contact line by the roughness features and a concomitant deformation of the interface with quadrupolar symmetry in the far field. Particles with near neutral intrinsic wettability (i.e. a contact angle near 90 degrees when the surface is smooth) can be dispersed either in the water phase or in the oil phase, and display increasingly large contact angle hysteresis with increasing surface roughness: upon interfacial adsorption the rough particles remain predominantly immersed in the liquid phase in which they were originally dispersed. The practically important implication is that roughness and order of wetting by the two liquid can be used to tune the particle contact angle in a wide range, and make the same rough particles suitable for the stabilization of both water- and oil-continuous emulsions.

The study is well-designed, its conclusions are important and well-supported by the data presented in the manuscript and the Supporting Information, and the manuscript is well-written; I therefore support its publication in Nature Communications without any significant revision.

One question the authors might want to comment on is the potential role of heterogeneous wetting, i.e. the entrapment within the rough surfaces of the liquid in which the particles are originally dispersed. It appears to me that the authors' analysis tacitly assumes even the meta-stable wetting with extreme contact angles (particles barely piercing the interface) to be homogenous (a single contact line, however corrugated, separating two solid-liquid interfaces). It is not clear to me that e.g. islands or films of entrapped water in the particle surface facing the oil would be detected experimentally if present, but they might provide an alternative explanation for the observed hysteresis. In this case the arguments made in the SI about the energy of pinning and dewetting would no longer be valid.

Reviewer #3 (Remarks to the Author):

The manuscript NCOMMS-17-02813-T by Dr. Isa and team presents a stunning study of the effect of particle roughness on the contact angle. This paper will surely attract much attention among diverse communities of researchers and especially those interested in wetting phenomena, particle stabilized emulsions, and physical chemistry of interfaces. In their comprehensive and logical work, Isa and co. prove that rough particles show completely different interfacial behavior compared to smooth particles. To the best of my knowledge, this has never been demonstrated experimentally as clearly as in this work. I believe, this finding is very important and inspiring for future fundamental research and applications of novel emulsifiers.

Their analysis, techniques, statistical validation, and reproducibility are all of high quality.

I highly recommend publication of this work, and only have one small comment/question:

The authors describe their strategy of smoothening by secondary TEOS precipitation on the surface of the PolyDADMAC and silica nanoparticle coated core silica particles. Can this TEOS coverage be incomplete, especially for the particles with the highest roughness? If so, that could result in different surface chemistries of the particles with different roughness. The particles with the highest roughness would for instance have a reduced amount of adsorbed didecyldimethylammonium bromide and consequentially a slightly less hydrophobic character than particles fully covered by silica. Can the authors proof that for all different roughness values, the surface is always completely composed of silica and the underlying PolyDADMAC does not contribute to the surface properties at all?

Martin F. Haase, Rowan University

The original comments appear in italics, while our response is in normal font. The corresponding revisions appear in red in the manuscript.

Reviewers' comments:

Reviewer #1 (Remarks to the Author):

The authors have shown how the surface roughness of a particle and not just its chemical composition can effect the adsorption of said particles at the oil/water interface. I find this manuscript well written and in my opinion the conclusions are well supported by their data.

However, I do not see the novelty of the work over and above that already published in the literature and in order to warrant acceptance into Nature Communications. Behrens and co-workers (ref 21) and Mable et al (ref 22) have already conducted extensive studies on how the surface roughness effects Pickering emulsifier performance. Therefore I cannot recommend publication.

We thank the referee for the positive comments on our work, but we firmly disagree with his/her statement about the novelty of our work in relation to the existing state of the art.

The two studies mentioned in the Reviewer's remark are duly cited in our article as two of the very few experimental attempts to link the surface topography of rough colloids to their performance as Pickering emulsion stabilizers. Crucially, in both studies the authors *characterize the particle morphology* (directly in the case of Mable et al., and more indirectly in the case of San Miguel et al.) *and link it to a macroscopic property of the emulsion*, i.e. adsorption efficiency (Mable et al.) or stability/maximum capillary pressure (San Miguel et al.). Before going into the details of the differences between our work and these two articles, we can summarize the novelty of our study as follows. Ours is the first set of experiments that connects a microscopic characterization of particle topography to interfacial adsorption at the level of individual colloids and to a macroscopic emulsification study unraveling a new class of emulsifiers. The ability to investigate *in situ* the behavior of single spherical rough microparticles using a range of complementary techniques is entirely new. Similarly new are the conclusions that we extract from these investigations and their consequences on the performance of rough particles as "universal" Pickering stabilizers.

In particular:

- The work of Mable et al. shows a versatile way to produce colloidal particles with finely tunable surface roughness in the form of tri-block-co-polymer vesicles by tuning the length of the various polymer blocks. The authors carried out a very careful characterization of the morphology of these particles by means of TEM and SAXS and then correlated the particle surface roughness to the adsorption efficiency upon emulsification. These

measurements look at how effectively the particles are trapped at the interface during emulsification by investigating the turbidity of the emulsion's continuous phase. *Their data therefore give "only" a macroscopic indication* of the percentage of adsorbed colloids, but have no means to investigate particle adsorption at the microscopic, single-particle level. The authors conclude that an optimum (or at least a minimum) surface roughness is needed to achieve high adsorption efficiency, but have no insights on the microscopic mechanisms for this finding.

- The work of San Miguel et al. represents a beautiful first experimental attempt to study of the effect of *particle surface roughness and link it to the macroscopic properties of an emulsion*; in this case, emulsion stability was tested by centrifugation. The main finding of this paper is a non-monotonic dependence of emulsion stability with surface roughness, which the authors linked to a wetting transition between a Wenzel and a Cassie-Baxter state. The authors came to this conclusion by looking at the *contact angle hysteresis* of water droplets *on macroscopic flat substrates* mimicking the particle surfaces. Both the contact angle measurements and the surface roughness characterization were done on macroscopic samples since the authors could not directly measure the adsorption and wetting of individual rough colloids at the interface. The latter task was, using their own words, "a formidable challenge". Our work directly tackles this challenge and shows that our level of microscopic analysis sheds new light on the adsorption of rough colloids at oil-water interfaces.

We hope that our extended comments have clarified our position in relation to the novelty of our work and we have specified even more clearly in the introduction of our manuscript the distinction between our work and preexisting literature.

Reviewer #2 (Remarks to the Author):

The manuscript by Zanini et al. describes a thorough study, in which model silica particles with tunable roughness and identical surface chemistry were used to evaluate roughness effects on the particle contact angle with and oil-water interface. The results confirm a previously predicted, but not experimentally confirmed pinning of the three phase contact line by the roughness features and a concomitant deformation of the interface with quadrupolar symmetry in the far field. Particles with near neutral intrinsic wettability (i.e. a contact angle near 90 degrees when the surface is smooth) can be dispersed either in the water phase or in the oil phase, and display increasingly large contact angle hysteresis with increasing surface roughness: upon interfacial adsorption the rough particles remain predominantly immersed in the liquid phase in which they were originally dispersed. The practically important implication is that roughness and order of wetting by the two liquid can be used to tune the particle contact angle in a wide range, and make the same rough particles suitable for the stabilization of both water- and oil-continuous emulsions.

The study is well-designed, its conclusions are important and well-supported by the data presented in the manuscript and the Supporting Information, and the manuscript is well-written; I therefore support its publication in Nature Communications without any significant revision.

We thank the reviewer for the very positive comments and for recommending publication without major revisions. Below we address the very pertinent point that he/she makes in relation to our observed contact angle hysteresis.

One question the authors might want to comment on is the potential role of heterogeneous wetting, i.e. the entrapment within the rough surfaces of the liquid in which the particles are originally dispersed. It appears to me that the authors' analysis tacitly assumes even the meta-stable wetting with extreme contact angles (particles barely piercing the interface) to be homogenous (a single contact line, however corrugated, separating two solid-liquid interfaces). It is not clear to me that e.g. islands or films of entrapped water in the particle surface facing the oil would be detected experimentally if present, but they might provide an alternative explanation for the observed hysteresis. In this case the arguments made in the SI about the energy of pinning and dewetting would no longer be valid.

The Reviewer makes a very good point, to which we reply as follows. We reply in a twofold manner. First, we can say that for the range of particles investigated in this study, even for the roughest ones showed in Figure 2aVI, which barely breach the interface and for which we can directly see the protrusion at the interface by FreSCa cryo-SEM, we do not detect isolated islands of water films exposed to the oil. We do see isolated asperities crossing the interface, but, in this case, the arguments made in the SI remain valid, as they indeed describe the dewetting of single asperities. The remark is nonetheless well taken and in the Discussion section, we have added a comment clarifying that our analysis is valid under homogeneous wetting conditions.

[Redaction]

We hope that the referee will find our clarification in the text and the evidence of heterogeneous wetting for other, rougher particles satisfactory in relation to his/her comment, and that he/she recommends publication of our revised manuscript.

Reviewer #3 (Remarks to the Author):

The manuscript NCOMMS-17-02813-T by Dr. Isa and team presents a stunning study of the effect of particle roughness on the contact angle. This paper will surely attract much attention among diverse communities of researchers and especially those interested in wetting phenomena, particle stabilized emulsions, and physical chemistry of interfaces. In their comprehensive and logical work, Isa and co. proof that rough particles show completely different interfacial behavior compared to smooth particles. To the best of my knowledge, this has never been demonstrated experimentally as clearly as in this work. I believe, this finding is very important and inspiring for future fundamental research and applications of novel emulsifiers.

Their analysis, techniques, statistical validation, and reproducibility are all of high quality.

I highly recommend publication of this work, and only have one small comment/question:

The authors describe their strategy of smoothening by secondary TEOS precipitation on the surface of the PolyDADMAC and silica nanoparticle coated core silica particles. Can this TEOS coverage be incomplete, especially for the particles with the highest roughness? If so, that could result in different surface chemistries of the particles with different roughness. The particles with the highest roughness would for instance have a reduced amount of adsorbed didecyldimethylammonium bromide and consequentially a slightly less hydrophobic character than particles fully covered by silica. Can the authors proof that for all different roughness values, the surface is always completely composed of silica and the underlying PolyDADMAC does not contribute to the surface properties at all?

We thank the Reviewer for the high praise of our work and the recommendation of publication. His/her comment is very pertinent and well taken. We provide here a detailed response to it and describe the corresponding revisions made to the manuscript.

The question only concerns the data presented in Figure 2a, where the particle surface is modified by the physisorption of didecyldimethylammonium bromide. It nonetheless does not concern the data in Figure 2b and 2c (and in the rest of the paper). The former particles were cleaned before the modification with bromo-silane to ensure an identical starting point for all surfaces (see Methods section), while the latter have no surfactant and are initially dispersed in the oil phase. Here, the tremendous difference in measured contact angles as a function of surface roughness cannot be ascribed to chemical differences between silica and PolyDADMAC, which are both hydrophilic, while the particles become trapped in a “highly hydrophobic metastable state” for the highest roughness.

The direct detection of residual PolyDADMAC exposed at the silica surface after smoothening is not a trivial task. We can nonetheless confidently state that topographical heterogeneity is mainly determining the behavior of the particles by carrying out electrophoretic mobility measurements on particles with different thickness of the smoothening layer. We have added additional data to Table 1 in the SI that show the estimated thickness of the smoothening silica layer for all the particles reported in Figure 2. This estimation is based on the complete

conversion of all the TEOS into silica considering the accessible surface area, also corrected for the roughness, of all the particles in solution during the smoothening process. The data reported in the graph below show the zeta potential of the rough particles corresponding to the data points IV (blue symbols) and VI (red symbols) in Figure 2a of the main text. These are the same raspberry-like particles on whose surface very different silica thicknesses have been grown, i.e. 60 and 10 nm, respectively. The filled symbols show the particle zeta potential versus pH for the native raspberries and the open symbols for the same particles modified by the physisorption of $C_{10}DMAB$ in equal conditions. Due to the difficulties of precisely linking electrophoretic mobility to zeta potential for rough particles (the standard model assumes uniform smooth spheres), small differences in the values of the former quantity can arise for different surfaces. The isoelectric point of the two systems will nonetheless be identical, if their surface chemistry is the same, and this is indeed observed in the figure by looking at the filled symbols. This is the same argument used by San Miguel et al. in Ref. 18 of the revised manuscript to ascertain that equal surface chemistries were obtained for their rough colloids. The modified particles additionally show an identical electrophoretic mobility over the whole pH range.

In addition to this, if we compare the data points III and IV in Figure 2a, these have significantly different silica thicknesses (20 and 60 nm, respectively) but show identical contact angles for very similar values of RMS roughness. Conversely, data points II and VI, have very similar, thin silica layers (8 and 10 nm, respectively), but show very large differences in contact angles, demonstrating that, even if the effect of residual exposed PolyDADMAC can play a role, this has a very minor effect compared to the role of surface roughness alone.

We thank the referee for raising this interesting point and we hope that he/she finds our response satisfactory and recommends publication of the revised manuscript.

REVIEWERS' COMMENTS:

Reviewer #2 (Remarks to the Author):

I appreciate the authors' response to my review (Reviewer 2) and recommend publication of the revised manuscript. Seeing that the discussion with the other two reviewers both referred to an earlier study of mine (San Miguel & Behrens, ref. 18), I will also reveal my identity and state for the record that I consider the authors' characterization of our work accurate.

Sven H. Behrens, Georgia Institute of Technology

Reviewer #3 (Remarks to the Author):

The reply by Dr. Isa and team is satisfying and I recommend publication of this accurate and extremely interesting work. They have carried out additional experimental work and included paragraphs in the manuscript addressing my points. The following describes my assessment of their reply:

The authors have shown by additional zeta potential measurement that for different silica smoothing thicknesses the isoelectric point (IEP) is identical. Although the IEP seems slightly elevated for pure silica (pH 4 instead of the typical silica value of pH 3) I believe that this is indeed a good proof for the uniform surface chemistry of the particles with different roughness used throughout the experiments.

Also their finding that particles with very different silica coating thicknesses but similar roughness showing equal contact angles, while particles having similar silica coating thickness but different roughness showing different contact angles highlights the roughness effect. Thus, I come to the conclusion that the main effect in their work is indeed the the surface roughness, while differences in surface chemistries might play only minor, negligible roles for the apparent contact angle.